# In-situ condition monitoring of floating offshore wind turbines using kurtosis and deep learning-based approaches.

Adrien Hirvoas[1], Cesar Aguilera[2], Matthieu Perrault[3], Damien Desbordes[3], and Romain Ribault[1]

[1]France Energies Marines, 525 Av. Alexis de Rochon, 29280 Plouzané
[2]SERCEL, 22 Rue des Platanes, 38120 Saint-Egrève
[3]SERCEL, 16 Rue du Bel air, 44470 Carquefou

**Correspondence:** Adrien Hirvoas (adrien.hirvoas@france-energies-marines.org)

**Abstract.** This study introduces a comparison between a kurtosis-analysis and a deep learning-based approach for condition monitoring of a floating offshore wind turbine. The study uses in-situ measurements from a 2.3 MW floating offshore wind turbine named Zefyros, deployed approximately 11 kilometers off the coast of Norway. The first method employs the statistical metric of kurtosis to detect unusual behaviors within a signal by identifying variations in the signal distribution. The second method employs a deep-learning procedure based on an auto-encoder approach, which transforms inputs into a reduced-dimensional latent space and then uses the encoded information to produce outputs identical to the inputs. One month of SCADA and high-frequency measurements obtained thanks to S-Morpho sensors were used in the study. Due to limitations in the accessible SCADA information, the anomaly scenario was simplified to detecting whether the turbine rotor was rotating or not. Both tested methodologies can accurately detect unwanted downtime periods, with the ground truth based on rotor rotations per minute (rpm) measurements. The auto-encoder method shows promising results, delivering more accurate outcomes than the kurtosis analysis on this in-situ measurement dataset. This study is a first step toward a more general use of auto-encoders for wind turbine condition monitoring. The latent space built by the auto-encoder can be leveraged to detect other types of unusual behavior, with a few labeled data.

## 1 Introduction

The Floating Offshore Wind Turbine (FOWT) is a rapidly expanding form of renewable energy. However, as of 2021, only 113 MW were operational in Europe (Ren21, 2021). The noise and visual pollution controversies surrounding onshore wind turbines, coupled with the potential for increased energy production, have prompted governments and institutions to propose ambitious plans for offshore sector expansion (WindEurope, 2022). FOWT systems, due to their minimized visual impact, are designed to be larger than their onshore equivalents (Gorostidi et al., 2023). This results in a larger swept area in areas where wind speeds are already high. These factors collectively contribute to a power output increase. Despite these advantages, the high costs associated with floating wind farms render them not yet competitive compared to fixed structures. In that context, the levelized cost of energy (LCOE) for floating offshore wind ranges between 90 euros/MWh and 120 euros/MWh (Gourvenec, 2020). In particular, approximately one-third of these costs are attributed to operation and maintenance (O&M) and other related activities (Nava et al., 2019). Consequently, the cost-effectiveness of wind turbines is negatively impacted by such high

O&M costs. In order to mitigate these costs, minimize unscheduled downtime, and ensure high availability of wind turbine assets, there is a significant demand for predictive maintenance tools and advanced anomaly detection methods.

Condition monitoring has been extensively researched across various domains and application areas. Numerous techniques have been developed for anomaly detection, falling into categories such as trending, clustering, normal behavior modeling, damage modeling, and assessment of alarms and expert systems (Tautz-Weinert and Watson, 2017). Especially in the wind
turbine field, several studies have explored the application of Supervisory Control and Data Acquisition (SCADA) for detecting anomalies (Stetco et al., 2019; Tautz-Weinert and Watson, 2017; Cui et al., 2018; Gonzalez et al., 2018; Stetco et al., 2019; Barahona et al., 2017). A variety of data-driven methodologies, such as neural network procedures (Zaher et al., 2009; Bangalore and Tjernberg, 2015), Support Vector Machine (SVM) (Dhiman et al., 2021), have been employed for wind turbine condition monitoring, all based on SCADA data.

However, in situations where SCADA data is either scarce or noisy, it may be difficult to construct a deep learning procedure for detecting anomalous behavior of the asset. This study explores the potential of using high-frequency in-situ data. In the following document, we delve into two distinct types of condition monitoring. Firstly, we examine a method that involves the manipulation and analysis of signals derived from magnetometers. This method aims to detect unusual behaviors by identifying variations from the standard geomagnetic field. Notably, this method employs the statistical metric of kurtosis to detect
anomalies within a signal by identifying variations in a distribution. Kurtosis provides a quantification of a distribution's shape by assessing the tails of the distribution in relation to its peak, thereby facilitating the recognition of outliers or unusual patterns in the data. Secondly, we suggest the application of deep learning techniques that utilize sophisticated neural network structures to autonomously learn and extract complex features from intricate data for the detection of anomalies within the data. In that context, autoencoders, a type of artificial neural network, have gained significant attention due to several advantages that are believed to be applicable to the wind energy sector. Primarily, autoencoders are used for unsupervised tasks, which
do not require labeled data during training. In more advanced configurations, autoencoders have proven to be efficient when used on semi-supervised problems (Tautz-Weinert and Watson, 2017; Berahmand et al., 2024; Sae-Ang et al., 2022), where only a limited number of labeled anomaly data are available for the training dataset. This is a notable advantage for wind turbines, as extensive historical failure data are typically not available. Furthermore, autoencoders can serve as a dimension-
ality reduction method and facilitate feature extraction from normal behavior data, which can subsequently be used for data clustering or classification (Berahmand et al., 2024). The proposed strategy for anomaly detection involves initially modeling normal behavior and subsequently leveraging this understanding to identify potential anomalies (Schlechtingen et al., 2013; Schlechtingen and Santos, 2014; Schröder et al., 2022). Most of the anomaly detection approaches have traditionally been applied to univariate time series data. However, as data becomes increasingly high-dimensional, detecting anomalies may re-
quire joint modeling of interactions between multiple variables. For such complex scenarios, deep learning approaches, such as convolutional autoencoders, offer distinct benefits. Convolutional autoencoders can capture intricate relationships and patterns within high-dimensional data, making them well-suited for anomaly detection tasks.

The remainder of the article is structured as follows. In Section 2, the mathematical description of the proposed methods for detecting anomalies is described. In Section 3, the detailed methodologies are applied to the in-situ data from an offshore wind

turbine operating in the North Sea. The resulting model performances in detecting operating and non-operating periods are shown based on a limited one-month dataset. In Section 4, conclusions of this study are drawn and future work is suggested.

## 2  Methodology

Condition monitoring involves first developing a procedure to recognize normal patterns and then generating anomaly scores to identify unusual behavior in assets. Due to limitations in in-situ measurements, identifying anomalies in the wind turbine's

behavior presented significant challenges. In this study, high-frequency sensors were deployed on the structure and used to detect the operating periods of a floating offshore wind turbine. In this document, we explore two distinct types of condition monitoring methodologies. However, such high-frequency data alone does not directly indicate the turbine's operational state. Furthermore, only a limited set of SCADA variables was available during the project, averaged over 10-minute intervals over a one-month period. These variables included nacelle wind speed (measured by a sonic anemometer), yaw angle, and rotor speed.

Despite their limitations, these SCADA variables were utilized to identify the hourly periods corresponding to the turbine's operating and non-operating phases. Based on these two datasets—high-frequency sensor data and corresponding operational status labels—the two proposed methods were trained in a semi-supervised manner to automatically infer the turbine's operational state. Firstly, we examine a method that involves manipulating and analyzing signals derived from magnetometers. This method detects unusual behavior by identifying variations from the standard geomagnetic field using kurtosis estimation.

Secondly, we propose the application of autoencoder architecture to autonomously learn and extract complex features from acceleration data for behavior detection. Figure 1 illustrates the global workflow of these two methodologies.

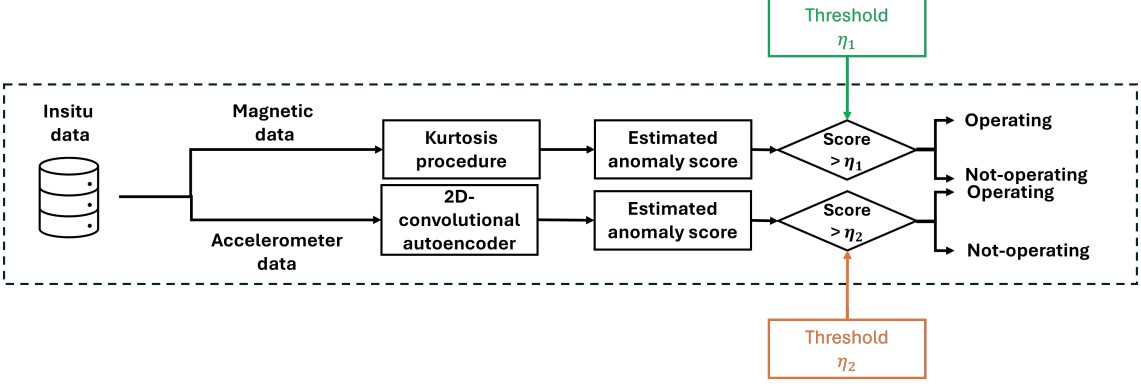

**Figure 1.** Workflow of condition monitoring methodologies. $\eta_1$ and $eta_2$ represent respectively the threshold from the kurtosis procedure and the 2D-convolutional autoencoder obtained from the training process.

The signal analysis conducted hereafter relies on a noise-based detection method (Zhao et al., 2021) and a kurtosis procedure. Indeed, one way to identify intermittency is to determine the probability density function (PDF), looking at how it differs from a Gaussian shape and especially for the presence of enhanced high-energy tails (Franco et al., 2021). In that context,

the approach relies on a statistical tool to analyze characteristics of data. This kurtosis indicator refers to the tailedness of a

distribution, and therefore, to the outlier occurrence. The kurtosis parameter, also known as the standardized fourth central moment of a distribution, can be defined as:

$$K(X) = \frac{\mathbb{E}[(X - \mu)^4]}{\sigma^4},$$

where $\mu$ and $\sigma$ are respectively the mean and standard deviation of the studied data denoted by $X$.

As mentioned in (Wang et al., 2016), kurtosis-based techniques are efficient methods for signal processing in the context of anomaly detection. In the wind turbine domain, the statistical approach is also used for the detection of bearing failures, see (Elforjani and Bechhoefer, 2018). In our application, to determine the wind turbine state, a magnetic signal obtained from a magnetometer located close to the rotor-nacelle-assembly (RNA) was analyzed. Such sensors measure the magnetic field for the three physical axes $(x, y, z)$ in $\mu$T (micro Tesla). This field is sensitive to electromagnetic interference from sources other

than Earth's magnetic force, such as the magnetic effects of electric currents or the presence of ferromagnetic materials.

     The deep learning method discussed hereafter relies on autoencoder models, which consist of two components: an encoder that maps input data to a low-dimensional space, and a decoder that reconstructs the original input data from this lower-dimensional representation. By structuring the problem this way, the encoder learns an efficient compression function, allowing the decoder to successfully reconstruct the original data. Autoencoders are commonly used for dimensionality reduction or

noise removal from images and data compression. The specific mapping learned by an autoencoder is tailored to the training data distribution, making it crucial for condition monitoring. When applying an autoencoder, the general principle is to first model normal behavior and then generate an anomaly score for each new data sample. This approach follows a semi-supervised paradigm, leveraging both large amounts of unlabeled data and small amounts of labeled data. The autoencoder is a specific architecture of artificial neural network (ANN), see (Chen and Guo, 2023). This mathematical concept relies on an artificial

neuron, which is a function of several variables that performs the linear combination of its inputs by arbitrary weights. Let us consider a $d$-dimensional parameter vector $\mathbf{x} = (x_1, \ldots, x_p)$ from the set $\mathcal{P}$ previously defined. A neuron operating on the vector $\mathbf{x}$ has $p$ weights denoted $\omega_1$ to $\omega_p$, representing the importance given to their respective input $x_j$. The output of the artificial neuron, noted $s \in \mathbb{R}$, is a real number obtained by weighted sum, such that:

$$s = \sum_{i=1}^{p} w_i x_i \cdot$$

This equation can also be represented in matrix form by the multiplication between the row vector $\boldsymbol{\omega}$ and the column vector $\mathbf{x}$:

$$s = \boldsymbol{\omega}^T \mathbf{x} = \begin{bmatrix} \omega_1 & \omega_2 & \ldots & \omega_p \end{bmatrix} \begin{bmatrix} x_1 \\ x_2 \\ \vdots \\ x_p \end{bmatrix}.$$

In general, an artificial neuron is equipped with a bias $b \in \mathbb{R}$ so that the modeled function is an affine function, that is, a linear function followed by a translation. This allows the neuron to not only scale inputs by weights, but also to shift (translate) the

function. This makes the model more flexible and capable of representing more complex patterns. It is defined as:

$$s = \boldsymbol{\omega}^T \mathbf{x} + b.$$

The neuron can also be equipped with a transfer function, also called an activation function, see Figure 2. We will denote this function as $\sigma : \mathbb{R} \to \mathbb{R}$. It transforms the weighted sum $s$ into an activation $z = \sigma(s)$. Therefore, the expression of the "complete" artificial neuron is:

$$y = \sigma\left(\boldsymbol{\omega}^T \mathbf{x} + b\right).$$

This equation represents a neuron that takes multiple inputs, applies a weight to each, sums them up, adds a bias, and then passes the result through an activation function to produce the final output. The activation function introduces non-linearity into the model, allowing the neural network to learn and represent more complex patterns. There are many different activation functions that can be used in artificial neurons, each with its own properties and use cases. The choice of activation function can significantly impact the model performance, e.g., linear, sigmoid, hyperbolic tangent, rectified linear unit, exponential linear

unit, softmax functions.

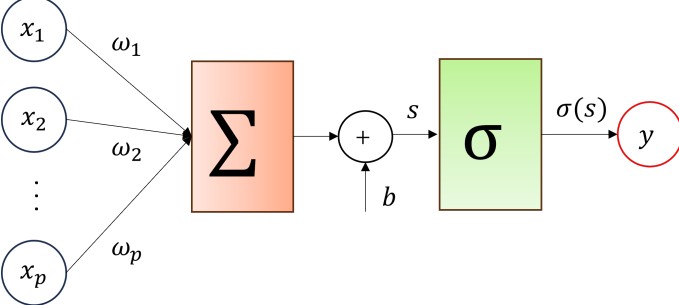

**Figure 2.** Graphical representation of a formal neuron. The neuron performs the weighted sum of the input activations, plus a bias. This quantity then passes through a an activation function $\Phi$.

Previously, we have presented the concept of a shallow network where only one hidden layer was considered. However, increasing the number of hidden layers in a neural network can enhance the model's ability to capture complex patterns and relationships in the data. This is because each layer in the network can learn to represent different levels of abstraction. Nevertheless, it is important to note that adding more layers also increases the risk of overfitting, where the model learns the

training data too well and performs poorly on unseen data. Therefore, the decision to increase the number of hidden layers should be made carefully, considering the complexity of the task and the amount of available data.

In that context, a feed-forward artificial neural network (FNN) is a set of formal neurons previously described which are organized in layers (Goodfellow et al., 2016). The autoencoder used in this paper relies on this specific feed-forward network architecture, allowing it to learn the most important features of the input. Mathematically, the $p$-dimensional input parameter vector, denoted by $\mathbf{x}$, is passed forward through $H$ hidden layers of neurons in order to compute the output vector $y$. Each

hidden layer consists of hidden neurons at which the incoming information is processed by the two steps previously described. Firstly, at each hidden neuron $j$ from the first hidden layer, the output $s^{(1)}$ is calculated by linearly scaling the input as follows:

$$s^{(1)} = \sum_{i=1}^{p} \omega_{ij}^{(1)} x_i + b_j^{(1)},$$

where $x_i$ is the $i$-th element of the input vector, $\omega_{ij}^{(1)}$ the weight connection between $x_i$ and the $j$-th hidden neuron from the first layer, and $b_j^{(1)}$ the bias offset for the $j$-th neuron. Secondly, the result is passed through an activation function $\sigma$:

$$y^{(1)} = \sigma(s^{(1)}).$$

By considering $\mathbf{X}$ as a random variable with values in $\mathbb{R}^p$ having a number of $H$ hidden neurons with a corresponding nonlinear activation function $\sigma_h$, we can define the complete FNN with the function $f_H : \mathbb{R}^p \to \mathbb{R}^{p_H}$ defined as:

$$f_H : \begin{cases} f_0(\mathbf{x}) = \mathbf{x} \\ f_h(\mathbf{x}) = \sigma_h(\mathbf{W}_h f_{h-1}(\mathbf{x}) + b_h), \quad \forall h \in [1, \ldots, H] \end{cases},$$

where, $\mathbf{W}_h \in \mathbb{R}^{p_h \times p_{h-1}}$ is the matrix containing the weights between layer $h-1$ and the layer $h$ and $b_h$ is the added bias to the layer $h$. In the training phase of an ANN, the goal is to estimate the weight parameters, denoted as $\mathbf{W}$, and bias parameters in such a way that the loss function result, i.e., the difference between the estimated output and the observed output, is minimized. This process can be achieved through the use of various optimization algorithms that iteratively find the parameters which minimize a selected loss function. The training process involves a technique known as back-propagation, which consists of two steps. The first step involves calculating the derivative of the loss function with respect to the weights. These derivatives are then utilized in the second step to update the model parameters. The term "epochs" refers to the number of iterations required to reach predefined stopping criteria. To evaluate the performance of the model, cross-validation (CV) can be used. This method estimates the generalization error by using different splits of training and testing data. It is evident that there are several hyper-parameters that need to be chosen in relation to the network structure (such as the number of layers and the number of neurons in each layer) and the training process (for example, weight initialization, learning rate, regularization factor, optimizer, etc.). The autoencoder, as illustrated in Figure 3, is composed of two interconnected FNNs. The first network, known as the encoder, reduces the data's dimensionality. The second network, referred to as the decoder, restores the data to its original dimension. The encoder and decoder have an equal number of layers and neurons, but their order is reversed. One of the strengths of autoencoder architectures lies in their adaptability to specific use cases. This adaptability is highlighted in (Berahmand et al., 2024), which provides detailed insights into how autoencoders can be customized for various applications. By considering $\mathbf{X}$ as a random variable with values in $\mathbb{R}^p$, an ANN, such as an autoencoder, models a function $\mathcal{H}$ such that:

$$\| \mathcal{H}(\mathbf{x}) - \mathbf{x} \| \leq \epsilon.$$

Mathematically, it means that the image of $\mathbf{x}$ is a reconstruction of $\mathbf{x}$ with an error of $\epsilon$. As said previously, to perform this reconstruction the autoencoder is decomposed into two parts:

      – an encoder, i.e., a function $\mathcal{E} : \mathbb{R}^p \to \mathbb{R}^d$,

     – a decoder, i.e., a function $\mathcal{D} : \mathbb{R}^d \to \mathbb{R}^p$.

By using this formalism, $\mathcal{H}$ is the successive application of encoding and then decoding, such that:

$$\mathcal{H} = \mathcal{D} \circ \mathcal{E}.$$

In principle, the encoder compresses the information by reducing the dimension of $\mathbf{x}$. So generally, the principle is to encode $\mathbf{x}$
with fewer variables than the input vector, such that $d \leq p$. Subsequently, we denote $\mathbf{z} = \mathcal{E}(\mathbf{x})$ as the vector of latent variables associated with $\mathbf{x}$. The objective of the training phase is to seek the weights and biases such that:

$$(\mathbf{W}, \mathbf{b})^* = \mathrm{argmin}_{(\mathbf{W}, \mathbf{b})} \mathcal{L}(\mathbf{x}, \overline{\mathbf{x}}),$$

where $\mathcal{L}$ is a regression cost function, e.g., mean squared error or mean absolute error.

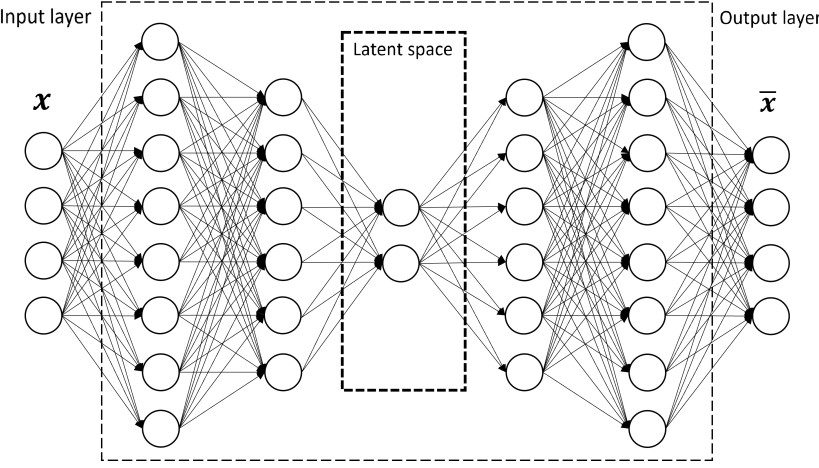

**Figure 3.** Graphical representation of an example autoencoder.

Previously, we focused on neural structures of the type known as FNNs within the autoencoder architecture. Nevertheless, these models have considerable practical limitations. Indeed, the use of fully connected layers is not suitable for all types of observations. In fact, the number of parameters in a multilayer perceptron grows rapidly with the dimension of the input and the dimension of the hidden layer. This explosion in the number of parameters poses two problems. First, computational costs increase, both in terms of computation time and the memory required to store the model's weights. Second, the massive increase in the number of parameters promotes overfitting and thus complicates decision modeling. In our research, we suggest utilizing convolutional autoencoders to identify and extract the most significant features from the input data. Such convolutional autoencoders leverage convolutional layers for feature extraction and an encoder-decoder architecture to learn compact representations of input data.

In this context, it is necessary to rely on the architecture of convolutional neural networks (CNNs) that allows for an a priori understanding of the spatial structure of the data. The fundamental operation in a CNN is the convolution operation between two function $f : \mathbb{R} \to \mathbb{R}$ and $g : \mathbb{R} \to \mathbb{R}$, defined as:

$$(f * g)(x) = \int\limits_{-\infty}^{+\infty} f(t)g(x - t)dt.$$

Even though the equation above is valid for univariate functions, it extends without difficulty to functions of several variables. In this case, the integral then covers the set of variables, i.e., multiple integral. Then, convolutional autoencoders (CAEs) are a specialized type of autoencoder that leverage layers with such convolutional operation (Lee and Carlberg, 2020). In the realm of signal processing, these convolutional autoencoders have been used to extract signal features (Wu et al., 2022). In summary, the ability of convolutional autoencoders to combine the strengths of autoencoders and convolutional neural networks makes them valuable tools for various signal processing tasks.

As said previously, the encoder compresses $\mathbf{x}$ by reducing its dimensionality while the decoder reconstructs $\mathbf{x}$ from the latent space $\mathbf{z}$. The space in which $\mathbf{z}$ evolves aims to be an underlying space that effectively explains the structure of the data $\mathbf{x}$. The concept is that significant differences in the observation space can be explained by small variations in the latent space due to underlying regularities in the distribution. In that context, this type of deep-learning architecture can be employed for anomaly detection by training the autoencoder with data from normal operating conditions (Chen et al., 2020; Renström et al., 2020). This training process enables the model to learn a mapping function that accurately reconstructs normal data samples with minimal reconstruction error. As a result, the neural network becomes proficient at reconstructing regular data but struggles when faced with data from abnormal or irregular operating conditions.

Finally, anomalies can be identified by examining the discrepancy between the input data and the reconstructed data. After fitting the autoencoder model, the next step involves determining an appropriate threshold for distinguishing between normal and anomalous sequences. One common approach is to set the threshold based on a quantile of the reconstruction error distribution. If the quantile is set high, more anomalous sequences will be classified as normal; conversely, a lower quantile will result in normal patterns being labeled as anomalous. As depicted in Figure 4, the choice of the threshold depends significantly on the specific study case. Based on the arbitrary value obtained from the training set, future examples can be classified as anomalous if the reconstruction error is higher. In this study, we have set the threshold at $\mu + 3\sigma$, where $\mu$ and $\sigma$ are the mean and standard deviation of the training reconstruction error, respectively.

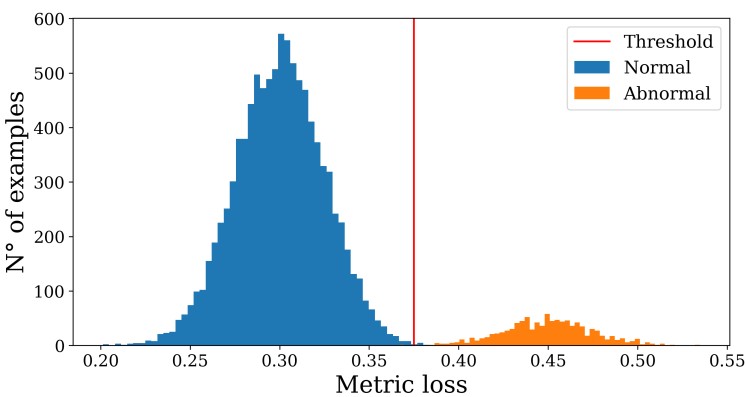

**Figure 4.** Representation of the losses distribution of an autoencoder for the non-anomalous detection.

## 3 A study case: the Zefyros floating offshore wind turbine

### 205  3.1 Study description and data preparation

Unitech Zefyros is a floating spar offshore wind turbine, originally installed as Hywind Demo by Equinor (formerly Statoil), located approximately 11 kilometers off the west coast of Karmøy, Norway (Ibrion and Nejad, 2023). The floater configuration is based on a vertically submerged cylindrical structure, as illustrated in Figure 5. This offshore floating system supports a Siemens 2.3-MW wind turbine with a rotor diameter of 82.4 meters. Table 3.1 contains key data for the SWT-2.3-82 wind

turbine and the floating platform structure.

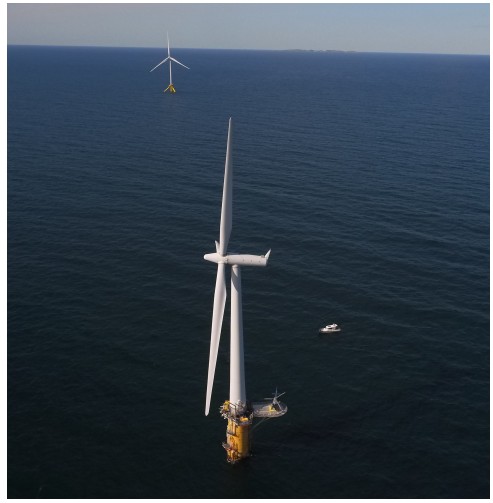

**Figure 5.** The Zefyros floating offshore wind turbine relies on a Siemens 2.3-MW. Credits: Unitech Energy Group

| Property | Value |
|---|---|
| Depth to platform base below SWL (m) | 100.045 |
| Elevation to platform top (tower base) above SWL (m) | 17.000 |
| Platform mass (kg) | 4,855,290.000 |
| CM location below SWL along platform centerline (m) | -74.770 |
| Hub height (m) | 65 |
| Hub mass (kg) | 26,400 |
| Rotor diameter (m) | 82.4 |

**Table 1.** Zefyros Properties

Due to the significant distance between the center of gravity and the center of buoyancy, the structure has demonstrated promising stability. The floating platform is known for its deep draft, which provides the buoy with excellent stability and hydrodynamic characteristics. The spar design limits wave-induced movements and loading on the structure by reducing the cross-sectional area in the splash zone, while simultaneously lowering substructure costs.

The Hywind Demo structure features a unique design where the wind turbine is supported by an underwater floating structure, similar to those used in offshore oil rigs (Liu et al., 2019). The Zefyros platform is ballast-stabilized and moored to the seabed using three separate catenary mooring lines. These mooring lines are anchored to the sea floor with drag anchors. Near the still-water sea level, each mooring line splits into a bridle attachment that connects to the hull, enhancing the system's yaw stiffness. Additionally, large clump weights are attached to each mooring line near the mid-span. The mooring lines are designed with chain and wire to achieve the desired force-displacement characteristics.

The success of the FOWT system is evident from its flawless operation over the years, demonstrating its ability to withstand various wind and wave conditions (Jacobsen and Godvik, 2020).

Zefyros's impact extends beyond its individual structure, as its design and the conception methodologies have influenced the design of the new structures for offshore wind turbines, contributing to the economic viability of offshore wind energy generation in deep waters Ibrion and Nejad (2023). Indeed, the first floating wind turbine's measurements and simulations have provided valuable insights into the performance and behavior of floating wind turbines, guiding further research and development in this promising technology field (Skaare et al., 2014).

| Node | Height (m) |
|------|-----------|
| 1 | 17 |
| 2 | 41 |
| 3 | 63 |
| 4 | 57 |
| 5 | 49 |
| 5 | 33 |

**Table 2.** Position of sensor nodes along the wind turbine tower

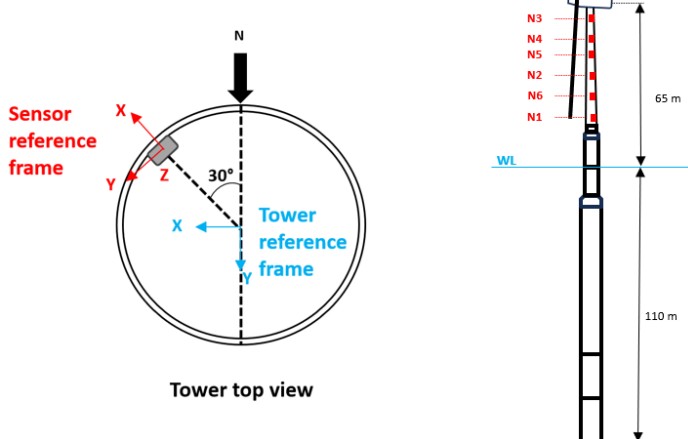

**Figure 6.** Projection of the sensor frame into the moving reference nacelle frame (left) Tower sensor location (right).

Hereafter, two strategies for identifying the operating phases of this wind turbine system using high-frequency sensor data have been developed. Both strategies are based on the acceleration and magnetic signal outputs from the S-Morpho sensor system. This system comprises six sensor nodes fixed along the tower, as shown in the left image of Figure 6. All sensor nodes

were installed approximately 30° west-north. The distances between nodes are specified in Table 2, with the mean sea level as the reference point, 17 meters below the tower bottom.

The first strategy is based on an engineering signal processing approach. The underlying methodology relies on the kurtosis analysis of a time signal from the magnetometer located near the rotor-nacelle assembly. One major advantage is that magnetometers are sensitive to electromagnetic interference from sources other than the Earth's magnetic field. In our context, permanent magnet generators in the wind turbine system generate a time-varying magnetic field. This variation can be detected using a magnetometer, which measures the strength and direction of magnetic fields.

In summary, by analyzing the acquired magnetic data from a magnetometer placed near the wind turbine generator, one can detect changes in the magnetic field patterns, providing valuable insights into the operational status of the generator.

The second strategy, based on an autoencoder mathematical approach, facilitates the detection of deviations from standard behavior established in a reference dataset. If necessary, the neural network architecture can identify the frequency range of unusual behaviors by examining the reconstruction error per frequency band. In our study, the neural network takes the power spectrum density (PSD) of sensor channels as input. The autoencoder process then recognizes common patterns in the spectrum. If these common patterns are not provided as input to the network, the model will struggle to reconstruct the PSD.

The in-situ data used to train and test the autoencoder in this study were based on measurements obtained from tri-axial accelerometers with an acquisition rate of 40 hertz, located inside the mast of the floating wind turbine, as illustrated in Figure 7. Their response in the frequency domain using the PSD has been studied for each one-hour acceleration response of the system. Besides the advantage of not using the full time series, the spectral response contains information on different physical phenomena, as illustrated in Figure 8, which can be identified using techniques such as operational modal analysis (Brincker and Ventura, 2015; Reynders, 2012).

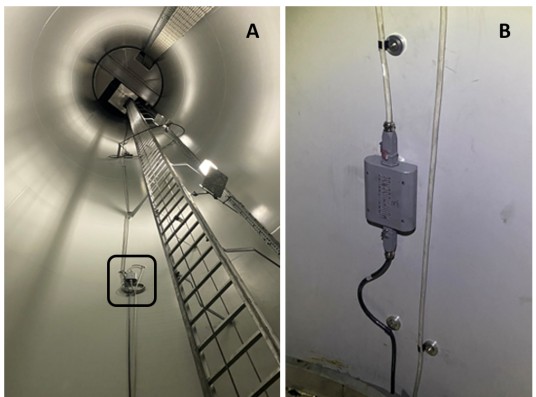
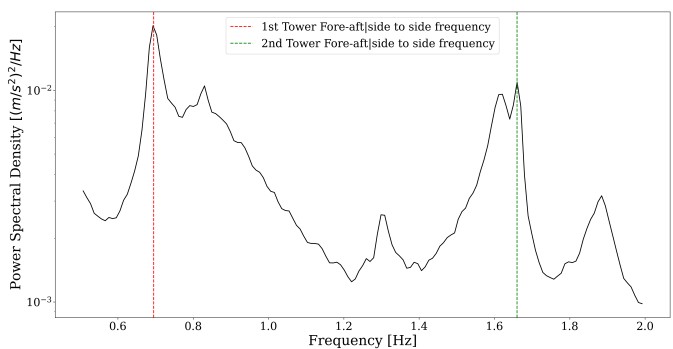

**Figure 7.** S-Morpho Installation in the Tower rounded in black on left figure.

**Figure 8.** Power spectral density of the tower acceleration measured by S-Morpho sensor obtained through the Welch method.

Due to some limitations in environmental condition measurements and system responses, challenges were encountered in identifying wind turbine anomalies. To address this, the data from tri-axial accelerometer sensors previously described were used to identify the operating period of the floating offshore wind turbine system using the aforementioned methods. However,

the acceleration data alone do not directly reveal the turbine's operating state. Additionally, during the project, we had access to a limited number of SCADA variables, which were averaged over 10-minute intervals for one month. These variables include wind speed at the nacelle, measured by a sonic anemometer, yaw angle, and rotor speed.

These limited SCADA system variables were used to identify the hourly periods corresponding to the wind turbine's operating and non-operating phases. Subsequently, a one-hour averaging and filtering process was applied to the raw data. The filtering method is based on the different operational regimes of a wind turbine, with a focus on the active operating phases (the area between the vertical dotted lines in Figure 9). When the wind speed exceeds the cut-in threshold, the turbine begins to rotate and the generator starts producing electricity. At the rated wind speed, the turbine reaches its nominal power output, constrained by the converter's capacity. Once this rated power is achieved, it must be regulated to prevent exceeding the generator's limits. Using this information, timestamps corresponding to operational and non-operational conditions were identified. These timestamps were then used to categorize the measured acceleration data into the same two groups, forming the input training dataset for the autoencoder. A schematic representation of the dataset selection process for the two approaches is shown in Figure 10.

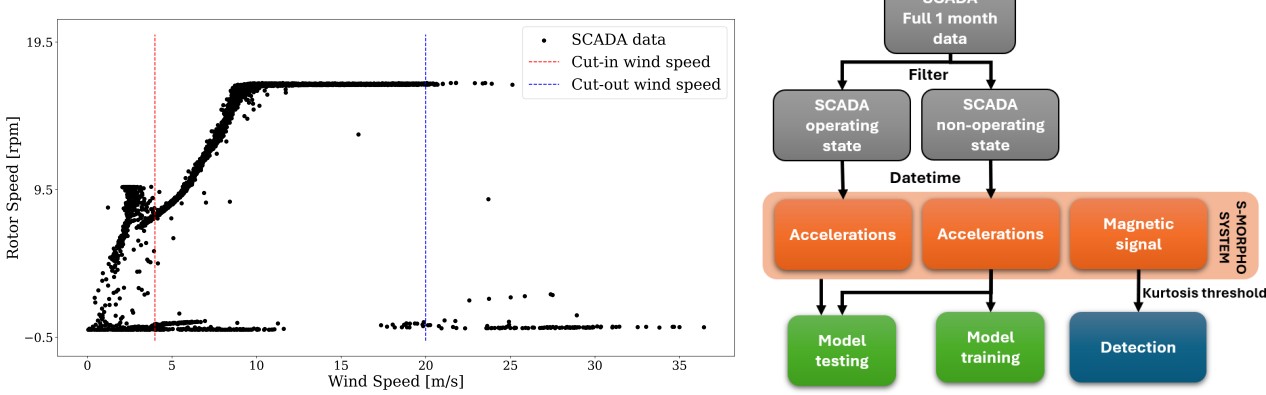

**Figure 9.** Averaged rotor speed vs. Averaged wind speed data over 10-minutes from SCADA system.

**Figure 10.** Graphical representation of the data set selection process.

## 3.2 Application and results

### 3.2.1 Signal processing approach

The analyzed magnetic signal was acquired by the 3-axis measurement system S-Morpho, which includes three accelerometers, three magnetometers, and one temperature sensor. As mentioned previously, only one sensor node was selected for this analysis campaign, specifically the closest sensor to the RNA. For more details on the location and orientation of the sensor nodes installed on the wind turbine tower, see (C. and M., 2024).

A spectral analysis of the magnetic time series was carried out as a pre-processing step to determine and visualize the frequency band of the rotating machinery inside the RNA. As shown in the spectrogram in Figure 11 B, a dominant frequency

emerged at around 11 Hz when the system operated at its rated condition. This frequency may be related to the high-speed shaft rotation inside the generator. In the image A in the same Figure 11, the rotation of the low-speed shaft (rotor) is illustrated as the black line in revolutions per minute, and the temporal series of the magnetic signal as the blue line. A significant change in magnitude can be seen around the 4500-second mark, indicating when the wind turbine rotation starts to slow down. This analyzed signal has a total length of two hours and was recorded on December 30, 2022, from 11 a.m. to 1 p.m. UTC.

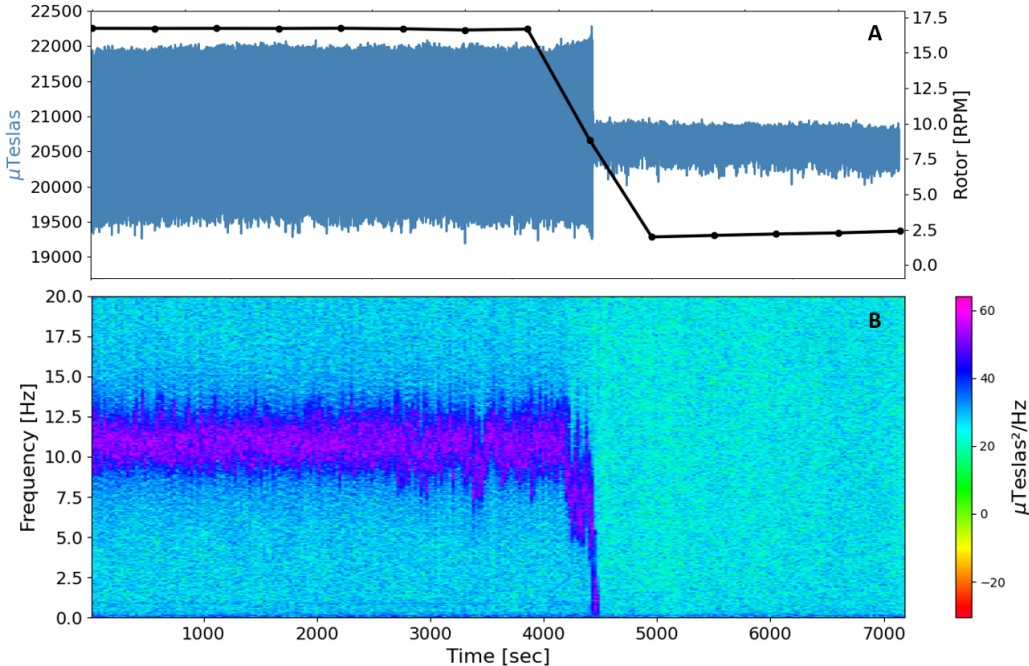

**Figure 11.** In the lower image, the magnetic time series (blue line) of the sensor number 3 along the Y axis. In the same image, the rotor rotation from collected SCADA data (black line). The spectral representation of the same magnetic signal is depicted in the upper figure.

It is important to note how the signal is affected by the rotating machinery, as seen in the spectrogram after the 4500-second mark. During this period, the signal seems to approach Gaussian noise as the low-speed shaft slowed down. With no energy production and no rotation, there is no magnetic interference detectable by the magnetometer sensor. Given this context, kurtosis analysis is an appropriate indicator, as it describes the characteristics of the dataset in terms of the distribution's tails based on a Gaussian distribution.

In this context, a detection methodology was developed to identify changes in the magnetic signal based on the wind turbine's operating state. Firstly, a one-day period covering three wind turbine conditions—at rated capacity (16 RPM), at half capacity (around 8 RPM), and at parked condition—was selected. The kurtosis indicator was estimated based on a one-hour time window to be consistent with the one-hour environmental data used for input data stratification. The kurtosis values were then compared to the SCADA values to determine the thresholds used later for detection criteria and binary classification. The threshold definition procedure, based on the selected period, was as follows:

– Selection of a reference period for the threshold definition (07/12/2022)

– Estimation of the kurtosis indicator based on 1 hour magnetic time signal sampled at 4O Hz

   – Manual definition of upper and lower threshold reference values

   – Estimation of the reference key metrics values precision and recall

   – Threshold parametric trial and error analysis to find the optimal thresholds values that ameliorate the reference key
     metrics (re-estimation of precision and recall metrics and compared to the reference ones)

During the comparison, two operating states were identified: the first when the turbine operates at rated capacity (around 16
RPM) and the second under-rated condition (rotations below 16 RPM). These variations in rotation affect kurtosis estimation,
causing the indicator values to fall between the three kurtosis categories: mesokurtic, platykurtic, and leptokurtic (Hatem et al.,
2022). This effect can be seen during the first days of the studied month, as shown in Figure 12. Thus, two kurtosis thresholds
were defined: a lower threshold at 2.8 and a higher threshold at 4.

Once the thresholds were defined, the magnetic temporal signal of December 2022 was analyzed. In Figure 12, the kurtosis
values of December 2022 are depicted as the orange line for a one-hour time window. It was possible to define kurtosis
thresholds since the magnetic field is sensitive to the wind turbine rotor rotation. When the rotor is not rotating, the acquired
magnetic signal is very close to Gaussian noise, and by definition, a kurtosis value based on Gaussian noise lies around 3 for
Pearson kurtosis. When the rotor rotates, it modifies the magnetic field, causing the kurtosis values to diverge from 3. In most
cases, when the wind turbine operates at rated condition, the kurtosis values decrease to around 1.8. Conversely, when the wind
turbine is rotating at a very low rate (0.3 to 1 RPM), the estimated kurtosis values increase to between 4 and 5. This is why it
was decided to set an upper and a lower thresholds.

   Then, the magnetic signal was classified in function of the thresholds as follows:

   – If kurtosis > 4 wind turbine operating state.

– If 2.8 <= kurtosis <= 4 wind turbine non-operating state (default detection)

   – If kurtosis < 2.8 wind turbine operating state

The two black dashed lines, in Figure 12, correspond to the selected threshold values, at 2.8 and 4. It can be seen that the
kurtosis values lies below the first threshold at 2.8 when the turbine is at rated condition, well depicted at the end of the month.
Often during the analyzed period, the rotor is not stood still, instead, a low rotor rotation is seen (December 2nd on Figure 12)
This low rotation has significant effects on the magnetic signal, which increases the kurtosis values. This is the main reason
why two thresholds were needed.

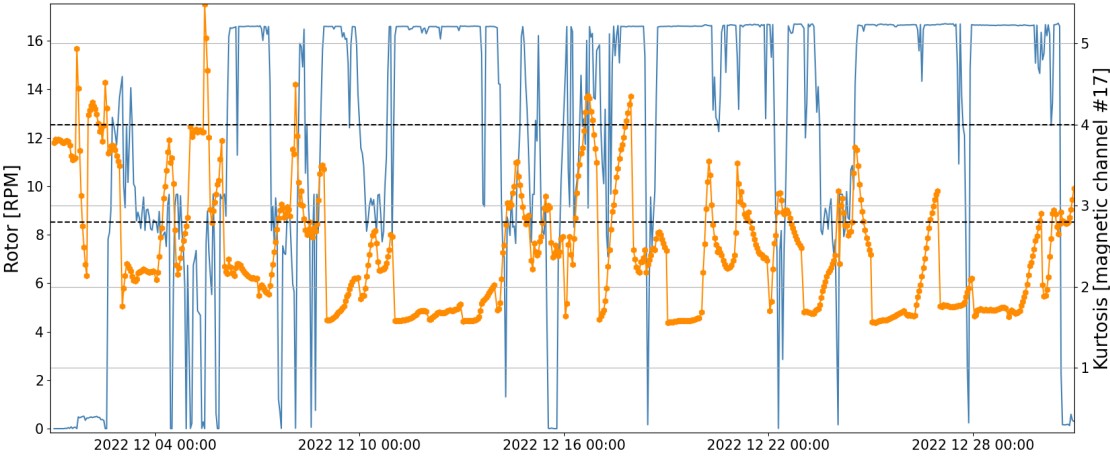

**Figure 12.** Temporal signal of SCADA rotor angular velocity as the blue line, and kurtosis estimation in orange. The two defined kurtosis thresholds are depicted as the two dashed lines.

### 3.2.2 Convolutional autoencoder approach

The accelerometers measure the vibration response of the wind turbine tower under varying environmental conditions, including different wind and wave intensities and directions. Given this context, the measurements are particularly sensitive to excitation properties. To ensure accurate condition monitoring, we trained a convolutional autoencoder using input samples that adequately cover the environmental condition space. If the autoencoder is not exposed to a diverse range of environmental conditions during training, it may incorrectly reconstruct the PSD response due to unseen excitations. One significant challenge we faced was defining an efficient validity domain of the model based on the environmental conditions present in the training dataset. To achieve this, we leveraged environmental conditions from weather numerical prediction models: ARPEGE for atmospheric variables (Déqué et al., 1994) and MFWAM for global ocean sea surface waves (Ifremer, 2025). Subsequently, we identified distinct training and testing sets by fitting a multi-dimensional histogram on the environmental variables and performing stratified sampling to select different one-hour periods for each wind turbine status.

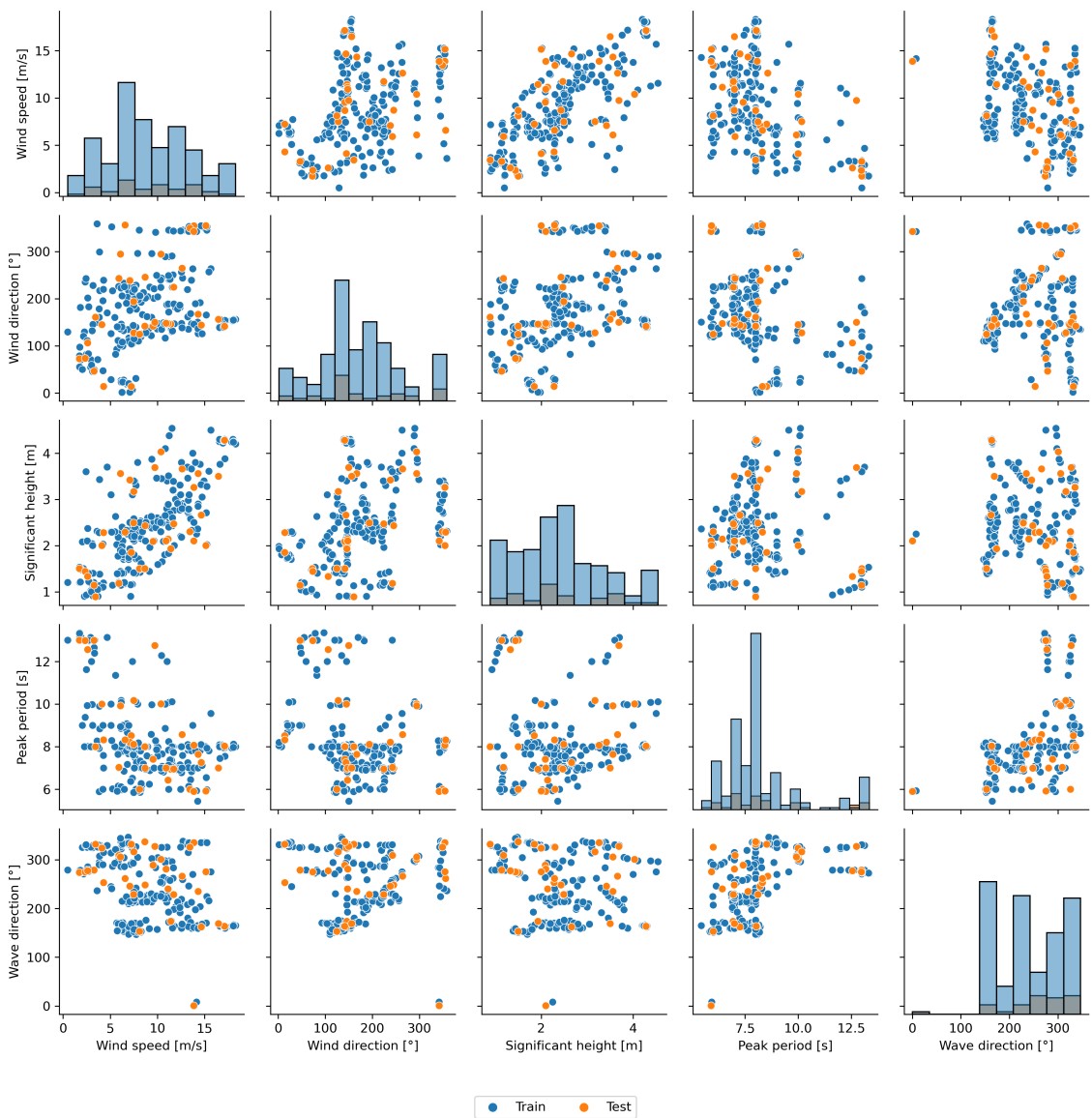

**Figure 13.** Stratified environmental input space on operating phase of the Zefyros floating offshore wind turbine to split the input data into train (blue dots) and test sets (orange dots).

As depicted in Figure 13, we thoroughly explored the entire environmental conditions space in both the training and testing datasets. By following this process, the autoencoder was trained on approximately 86% of the data and tested on the remaining 14% to prevent overfitting. As illustrated in Figure 14, two distinct sets of accelerations can be identified based on the wind turbine system's one-hour operating conditions. Ultimately, 257 operating hour periods with their associated accelerations

along the tower were identified in the raw data. Conversely, 24 non-operating statuses of the wind turbine were highlighted using the rotor speed from the SCADA system.

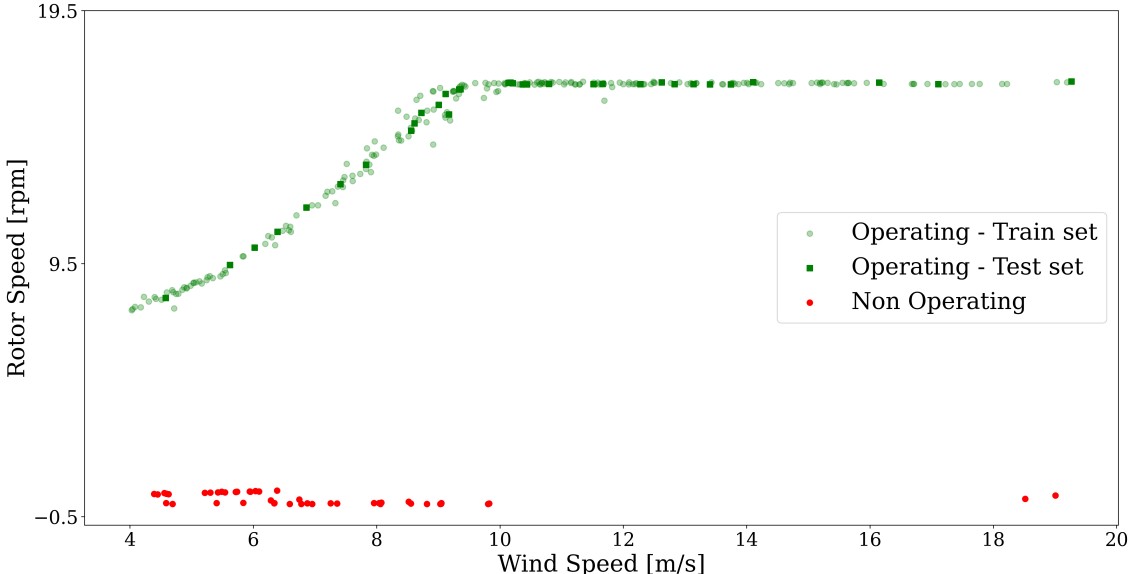

**Figure 14.** Rotor vs. wind speeds data from limited-SCADA with operating and non-operating 1-hour period.

The autoencoder under consideration is outlined in Table 3. A ReLU activation function is used between each layer, except
for the final layer, which employs a sigmoid function. During the training phase, the ANN architecture underwent parameter optimization using the Adam optimization algorithm, with a learning rate of 0.001 and a batch size of 5. The model was trained using back-propagation, with mean squared error serving as the loss function, defined as:

$$\mathcal{L}(\mathbf{x}, \overline{\mathbf{x}}) = \frac{1}{n_{signal}} \sum_{i=1}^{n_{signal}} (x_i - \overline{x}_i)^2 \,,$$

where $n_{\text{signal}}$ is the number of frequencies in the PSD. To detect when the wind turbine is operating, the reconstruction loss
produced by the autoencoder can be used to assess how well the signals are reproduced. Since the architecture has been trained on accelerations from operating periods, the model has learned the relationships between the signals under such conditions. As mentioned previously, the threshold is based on a quantile of the reconstruction error distribution. In this study, we have set the threshold at $\mu + 3\sigma$, where $\mu$ and $\sigma$ are the mean and standard deviation of the training reconstruction error, respectively. If the residual for new data exceeds the threshold obtained from the training error (see Figure 15), it potentially indicates a
non-operating status for the system.

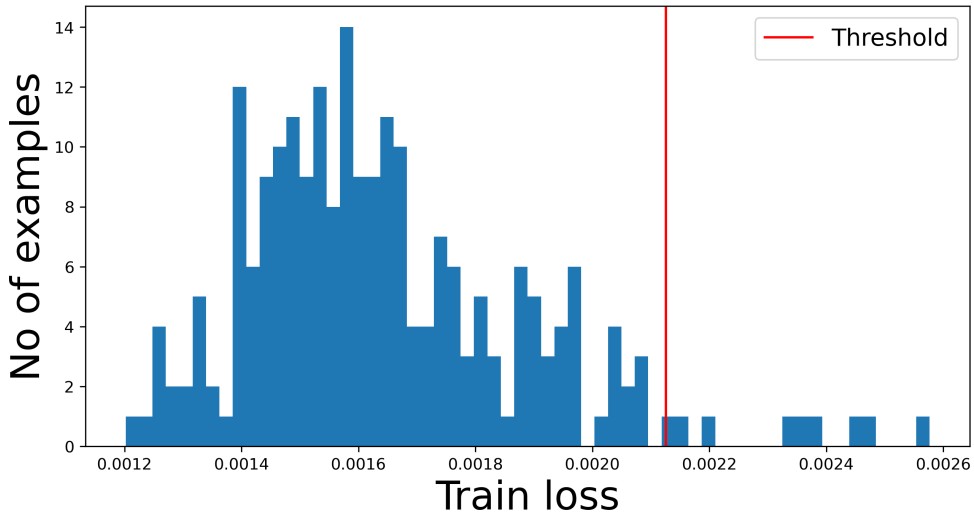

**Figure 15.** Representation of reconstruction error distribution applied to the training set on acceleration dataset from operating periods.

### 3.2.3 Results

The detection ability of the procedures has been tested on two distinct test sets: one dataset with identified operating periods and the other with non-operating behavior. In evaluating the performance of the condition monitoring procedure, key metrics such as precision and recall are usually computed. Precision, in the context of a binary classifier, quantifies the likelihood that a detected unusual behavior is indeed a true one. The mathematical definition of precision is as follows:

$$\text{Precision} = \frac{\#TruePositives}{\#TruePositives + \#FalsePositives}.$$

On the other hand, the recall metric, often referred to as sensitivity, estimates the system's proficiency in identifying unusual behaviors. The formal definition of recall is as follows:

$$\text{Recall} = \frac{\#TruePositives}{\#TruePositives + \#FalseNegatives}.$$

The calculated performance metrics for precision and recall on both detection test sets are summarized in Table 4. The proposed non-operating detection method, based on a convolutional ANN, demonstrates sufficient sensitivity in identifying wind turbine operating phases using tower accelerometer measurements. Notably, we achieved this without relying on feature engineering; instead, we allowed the autoencoder to learn spectral density patterns during the wind turbine's operational periods.

In Table 4, it can also be seen that the signal processing approach has lower performance than the convolutional ANN model. As a statistical method, the kurtosis value of a given distribution changes based on the sample size. When the sample is small, kurtosis values are highly variable. Therefore, the use of a large time window for kurtosis may significantly impact the detection method.

| Model | Architecture |
|---|---|
| Convolutional autoencoder | Input: $1 \times 18 \times 615$ (channels $\times$ height $\times$ width) |
| | Encoder: |
| | - Layer 1: 2D-Convolutional Layer (16, (2, 3), 2, 1) |
| | ReLu activation function |
| | Dropout layer with 0.1 rate |
| | - Layer 2: 2D-Convolutional Layer (32, (4, 3), 2, 1) |
| | ReLu activation function |
| | Dropout layer with 0.1 rate |
| | Decoder: |
| | - Layer 1: Transpose 2D-Convolutional Layer (32, (4, 3), 2, 1) |
| | ReLu activation function |
| | - Layer 2: Transpose 2D-Convolutional Layer (16, (2, 3), 2, 3) |
| | Sigmoid activation function |

**Table 3.** Architecture of the convolutional autoencoder with each convolutional layer defined as (number of channels produced, size of the kernel, stride of the convolution, padding added) and transposed convolutional layer defined as (number of channels produced, size of the kernel, stride of the convolution, padding added).

**Table 4.** Detection performance of the kurtosis analysis of a 1 hour time window and the condition monitoring method based on a convolutional ANN on a test dataset.

| | Kurtosis procedure | | Convolutional ANN | |
|---|---|---|---|---|
| | Operating | Not-Operating | Operating | Not-Operating |
| Recall | 0.939 | 0.750 | 0.970 | 0.917 |
| Precision | 1.000 | 1.000 | 1.000 | 1.000 |

To test the effect of the time window, a second analysis based on a 10-minute time window was performed. Similar to the first study, the data of the full month of December was used. The performance of this method was again compared to the SCADA data sampled at 10-minute intervals. From Table 5, it can be seen that performance has slightly diminished.

**Table 5.** Detection performance of the 10 min time window of kurtosis method.

| Test set | Precision | Recall |
|---|---|---|
| Operating | 0.99 | 0.887 |

As previously mentioned, the first method involves calculating the statistical metric of kurtosis. This mathematical approach quantifies the shape of the data distribution, particularly focusing on the tails relative to the peak, which helps in identifying outliers or unusual patterns. Under normal conditions, the kurtosis of the magnetometer signals remains within a certain range, but deviations from this range indicate potential unwanted behavior. By setting thresholds, this method can flag significant variations in kurtosis, thereby detecting anomalies related to unwanted downtime in our context.

The proposed deep-learning approach for detecting unwanted downtime periods in wind turbines employs autoencoders, a specialized type of neural network architecture, to produce outputs identical to their inputs. These models are trained in an unsupervised manner to learn optimal representations of operational scenarios. The novelty of this method lies in the use of a 2D-convolutional autoencoder, which is coupled with power spectra derived from on-site tower accelerations of the floating structure. This deep neural network architecture can accurately reconstruct inputs with similar characteristics, enabling the detection of anomalous patterns by identifying discrepancies between the input and reconstructed output, which may indicate potential downtime periods. The proposed methodology allows the autoencoder to autonomously learn and extract spectral features tailored to the components being monitored, eliminating the need for human intervention.

Both detection methodologies were validated using in-situ data collected from high-frequency sensors installed on the Zefyros floating offshore wind turbine. In conclusion, the deep-learning approach utilizing a 2D-convolutional autoencoder demonstrates superior performance in detecting unwanted downtime periods of wind turbines compared to the kurtosis signal analysis method. For example, Figure 16 depicts the accuracy of the convolutional autoencoder in identifying non-operational periods of the offshore floating wind turbine compared to the kurtosis method. The autoencoder's ability to learn complex patterns from power spectra derived from tower accelerations under various metocean and wind conditions enables more precise condition monitoring detection. Moreover, the analysis shows the high sensitivity of the kurtosis method's performance with the chosen time window. Consequently, this advanced deep-learning technique proves to be more effective and reliable, offering significant improvements over traditional kurtosis-based signal analysis for monitoring and maintaining wind turbine operations. The results obtained are highly encouraging, demonstrating the method's ability to successfully differentiate between operational and non-operational periods.

The comparative study highlights the advantages and limitations of both kurtosis analysis and deep learning-based approaches for condition monitoring in floating offshore wind turbines. While kurtosis analysis offers a straightforward statistical method for identifying deviations in magnetometer signals, its performance is highly sensitive to the chosen time window and may not be as reliable under varying operational conditions. On the other hand, the deep learning approach, specifically the 2D-convolutional autoencoder, demonstrates superior accuracy and robustness in detecting unwanted downtime periods. By leveraging power spectra derived from tower accelerations, the autoencoder can autonomously learn and extract complex patterns, providing a more precise and reliable condition monitoring mechanism. The validation results using in-situ data from the Zefyros floating offshore wind turbine underscore the effectiveness of the deep learning method, suggesting its potential for enhancing predictive maintenance and operational efficiency in the wind energy sector.

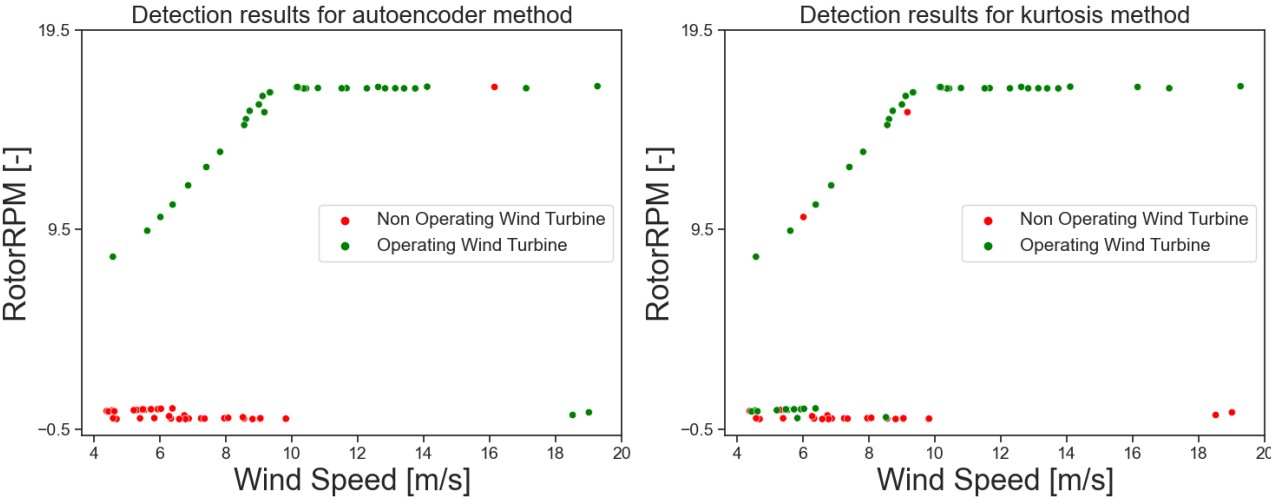

**Figure 16.** Rotor vs. wind speeds data from limited-SCADA with operating and non-operating 1-hour period from not seen data with the two methodologies.

## 4 Conclusions

Floating offshore wind energy is experiencing robust growth not only in Europe but globally. A significant portion of the levelized cost of energy is attributed to the operation and maintenance costs of these complex structures. The use of condition monitoring and artificial intelligence offers potent solutions for the automated early detection of abnormal wind turbine behavior under varying operational conditions.

The research work presented in this paper aimed to detail two condition monitoring procedures to detect unwanted downtime periods. The first method leverages the analysis of signals derived from magnetometers to identify unusual behaviors. Magnetometers measure the geomagnetic field, which typically exhibits stable patterns under normal operating conditions. By continuously acquiring and preprocessing these signals, we can monitor possible variations in the field. The principle of this method involves calculating the statistical metric of kurtosis. This mathematical solution quantifies the shape of the data distribution, particularly focusing on the tails relative to the peak, which aids in identifying outliers or unusual patterns. Under normal conditions, the kurtosis of the magnetometer signals remains within a certain range, but deviations from this range indicate potential unwanted behavior. By setting thresholds, this approach can flag significant variations in kurtosis, thus detecting anomalies related to unwanted downtime in our context. This approach provides a robust mechanism for real-time condition monitoring and can be integrated into the wind turbine's monitoring systems to enhance operational reliability and maintenance efficiency when SCADA is not available. The proposed deep-learning approach for detecting unwanted downtime periods in wind turbines employs autoencoders, a specialized type of neural network architecture, to produce outputs identical to their inputs. These models are trained in an unsupervised manner to learn optimal representations of operational scenarios. The novelty of this method lies in the use of a 2D-convolutional autoencoder, which is coupled with power spectra derived from

on-site tower accelerations of the floating structure. These spectra, similar to magnetic fields, are extracted from time series data spanning one hour and encompass a wide range of metocean and wind conditions. The autoencoder is trained using this input data, with the network's training dataset comprising operational data obtained through an analysis of SCADA variables. This deep neural network architecture can accurately reconstruct inputs with similar characteristics, enabling the detection of unusual behaviors by identifying discrepancies between the input and reconstructed output, which may indicate potential downtime periods. The proposed methodology enables the autoencoder to autonomously learn and extract spectral features tailored to the components being monitored, eliminating the need for human intervention.

Both detection methodologies were validated using in situ data collected from high-frequency sensors installed on the Zefyros floating offshore wind turbine. The deep-learning approach, utilizing a 2D-convolutional autoencoder, demonstrates superior performance in detecting unwanted downtime periods of wind turbines compared to the kurtosis signal analysis method. This comparison is substantiated by recall results from both procedures, which highlight the enhanced accuracy of the convolutional autoencoder, particularly in identifying non-operational periods of the offshore floating wind turbine. The autoencoder's ability to learn complex patterns from power spectra derived from tower accelerations under various metocean and wind conditions enables more precise unusual behaviors detection. Consequently, this advanced deep-learning technique proves to be more effective and reliable, offering significant improvements over traditional kurtosis-based signal analysis for monitoring and maintaining wind turbine operations. The results are highly encouraging, demonstrating the method's ability to successfully differentiate between operational and non-operational periods.

While the current study provides a promising start, there is scope for further investigations to enhance the condition monitoring procedure. Further studies are needed for applications beyond detecting the operational statuses of floating offshore wind turbines. The major challenge of the presented study was the limited access to SCADA data over a one-month period. Therefore, to detect and diagnose anomalies more effectively, it is necessary to gather more comprehensive information from the physical asset. Indeed, in the context of FOWT systems, it is valuable to explore and assess the developed damage detection methods across various damage types and intensities. By leveraging aero-servo-hydro-elastic simulations, we can investigate the system's response to different damage scenarios. Additionally, considering the temporal aspect is crucial when devising an effective condition monitoring strategy. As damage accumulates over time, understanding its progressive influence on detection accuracy becomes essential.

*Code availability.* The code used in this study is available upon request. Researchers and interested parties can obtain access by contacting the corresponding author. This approach ensures that the code is shared responsibly and that any questions regarding its use can be addressed directly. We are committed to promoting transparency and reproducibility in research, and we encourage collaboration and further exploration of our work.

*Author contributions.* This study on comparative anomaly detection for floating offshore wind turbines using in-situ data was a collaborative effort among all authors. Each author made significant contributions to the research and preparation of the manuscript. Adrien Hirvoas conceptualized the study, designed the methodology, and supervised the project. He also played a key role in data analysis, interpretation, and writing of the manuscript. Cesar Aguilera, Matthieu Perrault, and Damien Desbordes provided expertise in post-treatment of the in-situ data obtained from the accelerometers. They also reviewed and edited the manuscript for technical accuracy and clarity. Romain Ribault provided expertise in offshore wind turbine technology and in-situ data acquisition. He also reviewed and edited the manuscript for technical accuracy and clarity.

*Competing interests.* The authors declare that they have no known competing financial interests or personal relationships that could have appeared to influence the work reported in this paper.

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
