# Peer review of "In-situ condition monitoring of floating offshore wind turbines using kurtosis and deep learning-based approaches."

_Wind Energy Science, 2024_

## Referee Comment (RC2)

[referee-annotated manuscript omitted]

---

## Author Response (AR1)

**FEM comments**

**SERCEL comments**

I would like to express my gratitude to the anonymous reviewers for their careful reading, insightful comments, and valuable suggestions, which have significantly enhanced the quality and clarity of this work.

**Review 1**

Good manuscript overall, could use some improvements and expand on some sections, please see comments below:

- Line 135, not sure this sentence makes sense

  Sentence removed.

- On the methodology section, it would be useful to have a flowchart showing the two methods followed, which sensors they used etc

  Figure added with corresponding text

- Line 250: can you further elaborate on the filtering process? were there any curtailments to consider?

  The filtering process can indeed be further elaborated. While other filtering methods can be considered, it is important to note that there was restricted information available with the limited amount of SCADA data. Specifically, no additional information about curtailments was provided by the owner of the floating offshore wind turbine. Consequently, we decided to identify and remove data artefacts whenever necessary based directly on the SCADA. This step is crucial to ensure that the machine learning model is not biased by these artefacts.

- I understand that sensor location has been explained in another paper, but the authors could consider including a summary on their placement and sensor characteristics would be useful for the reader.

A schematic representation of the sensor nodes along the tower is depicted on the right side on figure below and the distances between nodes are specified on Table 1. As measurement reference point, the mean sea level  was taken as reference  , found it 17 m bellow the tower bottom.

Table 1: Position of sensor nodes along the wind turbine tower

| Node | Height (m) |
|------|------------|
| 1 | 17 |
| 2 | 41 |
| 3 | 63 |
| 4 | 57 |
| 5 | 49 |
| 6 | 33 |

[Figure]

In Figure X (a) is shown the local orientation of the sensors with reference to Noth direction. All sensors were installed approximatively 30° west-north. This configuration was chosen in order that local sensor x direction coincide with the Fore-aft movement of the nacelle reference frame, since the orientation selected corresponds to the most frequent wind direction. Therefore local sensor y direction corresponds to the Side-to-side and z direction to the upward and downward movement respectively.

- Figure 9: Further clarification on which sensor each signal comes from would and what processing was followed would be be useful.

  Add in caption Figure 10: "In the lower image, the magnetic time series (blue line) of the sensor number 3 along the Y axis. In the same image, the rotor rotation from collected SCADA data (black line). The spectral representation of the same magnetic signal is depicted in the upper figure."

- Line 275, how was the specific period of magnetic signal selected?

  In that context, a detection methodology was develop in order to identify the magnetic signal changes in function of the wind turbine operating state. Firstly, a 1 day period that covers three wind turbine conditions, at rated capacity (16RPM), at half capacity (around 8RPM) and at parked condition was selected. The kurtosis indicator was estimated based on a 1 hour time window period in order to be consistent with the 1 hour environmental data used as the input data stratification. Then, the kurtosis values were then compared to the SCADA values in order to determine the thresholds used later for the detection criteria and binary classification.

[Figure]

- Line 276: "The kurtosis values were then compared to SCADA values", can you elaborate in more detail how this comparison was done?

  Then, the kurtosis values were then compared to the SCADA values in order to determine the thresholds used later for the detection criteria and binary classification. The threshold definition procedure, based on the selected period was the following:
  - Selection of a reference period for the threshold definition (07/12/2022)
  - Estimation of the kurtosis indicator based on 1 hour magnetic time signal sampled at 40 Hz
  - Manual definition of upper and lower threshold reference values
  - Estimation of the reference key metrics values precision and recall.

- Threshold parametric trial and error analysis to find the optimal thresholds values that ameliorate the reference key metrics (re-estimation of precision and recall metrics and compared to the reference ones)

- Line 280: Can you explain or cite a source that explains the kurtosis categories?

@article{hatem2022normality,

 title={Normality testing methods and the importance of skewness and kurtosis in statistical analysis},

 author={Hatem, Georges and Zeidan, Joe and Goossens, Mathijs and Moreira, Carla},

 journal={BAU Journal-Science and Technology},

 volume={3},

 number={2},

 pages={7},

 year={2022}

}

- Line 285: can you further explain the connection between kurtosis thresholds and operating states?

In Fig. 10, the kurtosis values of December 2022 are depicted as the orange line for one-hour time window.
It was possible to define kurtosis thresholds since the magnetic field is sensitive to the wind turbine rotor rotation. When the rotor is not rotating, the acquired magnetic signal is very close to a Gaussian noise, and by definition, a kurtosis value based on a Gaussian noise lied around 3 for a Pearson kurtosis. When the rotor rotates, modifies the magnetic field and the kurtosis values diverges from the value of 3. In most cases, when the wind turbine operates in rated condition, the kurtosis values decreases at around 1.8. and regularly, when the wind turbine is rotating at very low rate (0.3 to 1 RPM) the estimated kurtosis values increases to values that lies between 4 and 5. This is why it was decided to fix a upper and lower threshold.

285 Then, the magnetic signal was classified in function of the thresholds as follows:

- Line 299: which weather numerical prediction models?

Information about weather numerical prediction models has been added.

- Line 308: Why were there two different activation functions used in different layers?

In certain cases, specific activation functions can be used to enforce properties on the output. For example, the sigmoid function returns values in the range of 0 to 1. This property is quite useful in our study due to the normalization of the quantity of interest. Therefore, we have decided to use the sigmoid function on the last layer to constrain the range of the predicted values from the artificial neural network.

- Figure 13: Can you comment on which data is beyond the threshold in the training phase? Were the non operating conditions used in the training?

As mentionned line 389, only operating conditions are used in the training phase. The 2D-convolutional autoencoder will learn optimal representations of operational scenarios.

- Line 333: Typo, "a statistical method"

Corrected.

- A bit more discussion on the results would be good. Do the authors recommend using ANN? Do they recommend bigger time windows for kurtosis estimation? did they do a sensitivity analysis on this?

Text added

- Also what type of anomalies can they detect with this methodology?

Due to limited access to the anomaly status of the wind turbine, the methodology has been validated to monitor operating and non-operating statuses. If new anomaly statuses become available, the methodology will be able to detect specific anomalies. Indeed, the semi-supervised aspect of this approach requires only a restricted amount of labeled data with corresponding anomalies.

- Additionally it would be nice to visualise the inference on the test/unseen data and how the model could detect anomalies in real time if implemented

A text with corresponding graphical representation has been added to visualize the inference on the test/unseen data. A GUI interface has been developed to automatically detect unwanted downtime period based on real-time measured accelerometers coupled with the developed autoencoder architecture. Nevertheless, this interface is not open-accessible.

- How common is it to not have SCADA over the sensors used in the paper? Are they widely available in commercial wind turbines? is this method more useful for field trials?

SCADA (Supervisory Control and Data Acquisition) systems are commonly used in the wind turbine field to monitor these systems via sensors deployed in real-world scenarios. However, high-frequency sensors, as presented in this paper, coupled with deep neural networks, seem to pave the way for anomaly detection without the need for feature engineering during post-processing. One major challenge will be to have a well-described training dataset with well-defined anomalies due to the semi-supervised nature of the approach.

**Review 2**

The article aims to detect abnormalities for the maintenance of an FOWT using two different methodologies. However, the article fails to articulate the methodology, data collection, results, and discussion. I attached my comment on the pre-print of the article. However, these are the general points that need to be addressed:

- The article's methodology section provides an in-depth overview of the basics of ANN, without being specific to the research presented.

We appreciate your feedback. We have restructured the methodology section to focus more specifically on the detection application to our research. Detailed steps, specific

configurations, and parameters used in our study will be included to enhance clarity and relevance to our research objectives.

- The statistical method needs a bit more explanation.
  In the revision 1 the following point have been added
    - a more detailed description about sensor placing
    - criteria for the selection of the signal used to determine thresholds
    - how thresholds were defined
    - More detail explanation between kurtosis values and wind turbine operating states
  - How are you sure that the normal behavior distribution is Gaussian?
    Two main indicators, the first is because when the wind turbine is not operating, the histograms of the magnetic signal are close to a normal distribution.
    Secondly, the kurtosis value of the signal is close to 3, implying normal /gaussian distribution.
  - This is a picture form tower internal. Where is the S-Morpho installation?
    Inside the figure was highlighted the sensor location. Another image was added, with a closer view of the sensor.
  - This is not a clear figure. Please use sub-figures a and b to refer to the sub-figures. What is the x axis in the bottom sub-figure? So, it seems you stop generating power, but the rotor is still rotating, why?
    X axis in lower figure are samples at 40 Hz. The figure was re-plotted using A and B subfigure indicators.
    Yes rotor is rotating at low speed and during the one month analyzed signal, this wind turbine is rarely stand still ( 0 RPM), mostly rotating at low speed. This was the main reason why we have used 2 thresholds, because this low rotation has impact on the magnetic signal,
  - Low speed shaft RPM is not at zero!
  - In your methodology, you need to explain kurtosis a bit more and tell the readers what are the three categories, and what do they mean. It is confusing for the readers.
    A reference was added where kurtosis categories are explained. They represents how the distribution diverges from a normal distribution. Platikurtic if distribution is more "flat" and Leptokurtic if distribution is more "pointy".
  - This figure needs discussion.
    More explanation was provided.
  - So, to this point you calculated Kurtosis for the magnetic channel, which is higher resolution, and then show how it behaves in comparison with RPM. How did you end up at the thresholds? What is the value of this detection?
    Explained in first revision:

    The threshold definition procedure, based on the selected period was the following:
      - Selection of a reference period for the threshold definition (07/12/2022)
      - Estimation of the kurtosis indicator based on 1 hour magnetic time signal sampled at 4O Hz
      - Manual first definition of upper and lower threshold reference values
      - Estimation of the reference key metrics values (precision and recall).
      - Threshold parametric trial and error analysis to find the optimal thresholds values that ameliorate the reference key metrics (re-estimation of precision and recall metrics and compared to the reference ones)

- The structure of the paper requires revision. I suggest having a separate section about the data and explaining what it what.

We have added information for data description, detailing the sources, types, and preprocessing steps involved in the sub-section called "Study description and data preparation". This new information provides an explanation of the data used in our study, enhancing the overall organization and readability of the paper.

- The thresholds and the methods to achieve them are not clear. They seem arbitrary.

We acknowledge the need for clarity regarding thresholds. We have provided a new explanation of how thresholds were determined for the two different approaches. This new information involves a justification for the chosen thresholds to ensure clarity.

- There is no discussion on the results.

Some discussion on the results has been added.

- The paper claims "anomaly detection," while it is only detecting if the turbine is operational or not. This is not an abnormality detection.

We have adjusted our terminology to accurately reflect the scope of our monitoring capabilities. The focus in the article is on operational status detection. The new terminology used throughout the paper is consistent and accurately represents our research objectives.

- The grammar and writing need to be improved.

We have thoroughly reviewed the manuscript for grammatical errors and improved the overall writing quality to ensure clarity and professionalism.

- It is important to publish research, but what I am missing here is the novelty of this work. How does it contribute to the community?

This research work is preliminary study allowing to highlight the potential of the deep-learning procedure linked to high-frequency sensors. Nevertheless, there is scope for further investigations to enhance the condition monitoring procedure. Future studies should focus on applications beyond detecting the operational statuses of floating offshore wind turbines. Gathering more comprehensive SCADA data over extended periods will improve the detection and diagnosis of anomalies. Additionally, exploring and assessing developed damage detection methods across various damage types and intensities will be valuable.

- Publishing the code and possibly the data helps the reader.

We agree that transparency is crucial. However, due to industrial confidentiality, we are unable to publish the code and data publicly. We can share the code and data upon request, ensuring that interested readers can replicate and build upon our work while maintaining confidentiality agreements.